# Effects of (*S*)-Carvone and Gibberellin on Sugar Accumulation in Potatoes during Low Temperature Storage

**DOI:** 10.3390/molecules23123118

**Published:** 2018-11-28

**Authors:** Yajing Xie, Jakaria Chowdhury Onik, Xiaojia Hu, Yuquan Duan, Qiong Lin

**Affiliations:** Institute of Food Science and Technology, Chinese Academy of Agricultural Sciences/Key Laboratory of Agro-Products Quality and Safety Control in Storage and Transport Process, Ministry of Agricultrue, Beijing 100193, China; xieyajing@caas.cn (Y.X.); j.conik@yahoo.com (J.C.O.); huxiaojia2009@foxmail.com (X.H.)

**Keywords:** cold-induced sweetening, potato, gibberellin, (*S*)-carvone, postharvest storage

## Abstract

Potato tubers (*Solanum tuberosum* L.) are usually stored at low temperature, which can suppress sprouting and control the occurrence of diseases. However, low temperatures lead potatoes to easily suffer from cold-induced sweetening (CIS), which has a negative effect on food processing. The aim of this research was to investigate potential treatments on controlling CIS in potatoes during postharvest storage. “Atlantic” potatoes were treated with gibberellin and (*S*)-carvone, respectively, and stored at 4 °C for 90 days. The results showed that gibberellin can significantly accelerate sprouting and sugar accumulation by regulating expressions of *ADP-glucose pyrophosphorylase* (*AGPase*), *granule-bound starch synthase* (*GBSS*), *β-amylase* (*BAM1/2*), *UDP-glucose pyrophosphorylase* (*UGPase*) and *invertase inhibitor* (*INH1/2*) genes. The opposite effects were found in the (*S*)-carvone treatment group, where CIS was inhibited by modulation of the expressions of *GBSS* and *INH1/2* genes. In summary, gibberellin treatment can promote sugar accumulation while (*S*)-carvone treatment has some effects on alleviating sugar accumulation. Thus, (*S*)-carvone can be considered as a potential inhibitor of some of the sugars which are vital in controlling CIS in potatoes. However, the chemical concentration, treatment time, and also the treatment method needs to be optimized before industrial application.

## 1. Introduction

Potato (*Solanum tuberosum* L.) is widely consumed as a staple food and snack. It contains plenty of nutrients and plays a crucial role in the food processing industry. To avoid the occurrence of sprouting and rotting, potato tubers are usually stored at low temperature. In potatoes, dormancy is the physiological state of the tuber in which autonomous sprout growth will be retarded by up to two weeks, even under suitable sprout growing conditions [1]. Several factors contribute to potato tuber dormancy, including variety, maturity status of the tubers at harvest, the tuber growth environment, post-harvest storage conditions, injury and exposure to endogenous and applied dormancy-breaking compounds [2]. Of these factors, genetic control (variety) may be the most significant [3]. Although low temperatures during storage can prolong the dormancy period, they generally result in an increase in reducing sugar content, primarily glucose, which is undesirable in the processing industry due to darkening of fried products [2]. This process is known as cold-induced sweetening (CIS). It has been reported that CIS caused unpleasant dark coloration and unacceptable bitter flavors due to Maillard reactions during the high temperature frying process [4]. It can also promote the formation of acrylamide, identified as “carcinogenic to humans” by the World Health Organization and International Agency for Research on Cancer [5].

CIS is a result of imbalance between starch and sugar metabolism at low temperature, with an increasing number of starches being converted into sugars. It is a complex process occurring with the participation if a variety of enzymes, such as starch phosphorylase (SP) [6], ADP-glucose pyrophosphorylase (AGPase) [7], granule-bound starch synthase (GBSS) [8], UDP-glucose pyrophosphorylase (UGPase) [9], sucrose phosphate synthase (SPS), sucrose synthase (SuSy) [10], α-amylase (Amy) and β-amylase (BAM) [11,12]. In addition, a strong link between vacuolar invertase (INV) and reducing sugars was observed, which is a catalyzed process degrading sucrose into fructose and glucose and negatively regulated by invertase inhibitor (INH) [13]. The sugar metabolism process has been studied for a long time, but the regulation of genes related to sugar metabolism under different storage conditions remains poorly understood. Several treatments were stated to be effective on controlling CIS in potatoes according to previous studies, such as low oxygen level storage [14], heat treatment [15], UV-C treatment [16] and 1-MCP treatment [17]. Gibberellin is a plant hormone which has been reported to play an important role in sucrose synthesis and abiotic stress responses [18]. A previous study on barley indicated that gibberellin can significantly up-regulate *amylase* expression, which causes the conversion of starch into soluble sugars and accelerated sweetening [19]. However, it is still unknown whether gibberellin has a similar effect on CIS modulation in potato tubers during storage. Plant essential oils are well-known as Generally Recognized As Safe (GRAS) compounds commonly applied as a natural alternative in food preservation [20], among which, a volatile monoterpene—(*S*)-carvone—which is successfully used to protect potatoes from wound healing, sprouting and fungal diseases, is of great interest [21,22,23]. However, few studies are focused on a possible relationship between (*S*)-carvone and CIS.

In the present study, the effects of gibberellin and (*S*)-carvone on sprouting, sugar contents, and CIS-related genes were evaluated. The aim of this study was to provide an alternative way, gibberellin treatment and (*S*)-carvone treatment, to control CIS and maintain the physiological qualities in potatoes during postharvest storage and explore its regulation method by quantitative real-time PCR (qRT-PCR) in a molecular level.

## 2. Results

### 2.1. Tuber Sprouting

Potato dormancy started to be broken from 30 days of storage in the gibberellin treatment group, while that in the control and (*S*)-carvone groups was delayed up to 60 days of storage (Figure 1).

Compared with potatoes in the control group, the sprouting rate increased significantly in gibberellin-treated potatoes at 30 days and 60 days of storage, and that was significantly decreased by (*S*)-carvone treatment at 90 days of storage. At 90 days of storage, the sprouts in gibberellin-treated potatoes grew faster and longer than the control, but changed only slightly with (*S*)-carvone treatment (Figure 2).

### 2.2. Physiological Response

Continuous weight loss was exhibited after all the treatments during the whole storage period, amounting to up to 1.13%, 1.39% and 1.37% by the end of storage in the control, gibberellin and (*S*)-carvone treated potatoes, respectively.

Thus, no significant difference was observed between the three treatments (Figure 3A). The respiration rate changed dramatically at the beginning of the storage period, followed by a steady trend later. No significant difference was found between treated samples and CK during the whole storage (Figure 3B).

### 2.3. Sugar and Starch Contents

Three predominant soluble sugars (sucrose, glucose and fructose) were measured in the present research. The sugar contents mostly increased during the whole storage period in all the treatment groups, where gibberellin-treated potatoes maintained a higher sugar content (Figure 4). In contrast to the control, gibberellin treatment greatly enhanced the sucrose content (Figure 4A) up to 30 days which was inhibited by the (*S*)-carvone treatment. At 60 days of storage, (*S*)-carvone-treated tubers showed a similar slope as the control while a significantly higher sucrose content was maintained in gibberellin-treated tubers. The glucose (Figure 4B), fructose (Figure 4C) and total reducing sugars (Figure 4D) were significantly increased from 60 days to 90 days of storage in gibberellin-treated samples compared to control and (*S*)-carvone treatment. However, significant differences were found between the control and (*S*)-carvone-treated tubers which inhibited the accumulation of glucose, fructose and total reducing sugars up to 60 days of storage indicating a dominant effect of (*S*)-carvone treatment to impair the formation of reducing sugars. Thus, an ultimate decrease of 7.13 g kg^−1^ and an increase of 14.53 g kg^−1^ of total reducing sugars were observed at 60 days of storage in (*S*)-carvone-treated tubers and gibberellin-treated tubers, respectively.

The total sugar contents (Figure 4E) were decreasing in all the treatment during the whole storage period, however, no significant difference was observed among the treatment.

### 2.4. Expression Patterns of Starch and Sugar Metabolism Related Genes

Involvement of several genes was indicated in our study, which had fluctuating effects under different postharvest storage treatments, indicating their roles in the accumulation of reducing sugars through the metabolism of starch and sugar. The expression of *SP* (Figure 5A), *AGPase* (Figure 5B) and *GBSS* (Figure 5C) were dominant immediately after storage up to 30 days and significant lower expressions of *AGPsae* and *GBSS* were observed in gibberellin-treated tubers over control tubers at 60 days and 90 days of storage, respectively, whereas (*S*)-carvone treatment enhanced the expression of *GBSS* at 60 days of storage. *SP* was also regulated significantly by increasing the expression at 60 days and 90 days of storage through (*S*)-carvone treatment. However, the expression of *SP* was not significantly affected by gibberellin treatment.

In contrast to the control tubers, gibberellin treatment momentously increased the *BAM1* expression (Figure 6A) at 30 days and *BAM2* (Figure 6B) expression at 60 days and 90 days of storage. In the case of *Amy23* (Figure 6C), a significant lower expression was observed during the whole storage period in gibberellin treated tubers. On the other hand, (*S*)-carvone treatment did not affect the expressions of *BAM1* and *BAM2* and the only significant difference was found in the expression of Amy23 at 60 days of storage when compared to control.

The expression of *UGPase* (Figure 7A) and *SPS* (Figure 7B) increased immediately after storage at 30 days whereas *SuSy* (Figure 7C) declined slightly. Profound variation was observed at 30 days of storage where gibberellin treatment enhanced the expression of *UGPase* compared to the control. 

The expressions of both *UGPase* and *SPS* were also higher at 30 days of storage and declined slightly up to 90 days in (*S*)-carvone-treated tubers. Although a slight diminution of expression in the early storage period was observed in *SuSy* and it was increased modestly from 60 days to 90 days of storage, the difference was not significant in both of the treatments compared to control.

*INV* (Figure 8A) showed an increasing expression up to 30 days of storage but higher expression in gibberellin-treated tubers only showed a significant difference with control and (*S*)-carvone treatment at 90 days of storage. *INH1* (Figure 8B) and *INH2* (Figure 8C) genes exhibited increasing trend up to 60 days of storage and the lowest expression was observed at 90 days of storage in gibberellin-treated tubers. Moreover, significant increase in expressions of *INH1* and *INH2* were observed at 30 days and 60 days of storage separately, following with a significant decrease in *INH2* at 90 days in (*S*)-carvone treated tubers compared to the control.

From the above results, gibberellin treatment up-regulated *BAM1* expression significantly at 30 days of storage and turned to affect expression of *BAM2* after 30 days of storage, which leads to a high starch degradation rate in gibberellin treatment. Meanwhile, gibberellin also significantly down-regulated genes related to starch resynthesis, like *AGPase* and *GBSS*, to suppress the accumulation of starch. The imbalance between starch degradation and resynthesis may cause the potato sweetening. In addition, gibberellin can significantly up-regulate *UGPase* expression within the first 30 days of storage, causing the accumulation of sucrose. Importantly, the expression of *INH1/2* were down-regulated significantly at 90 days of storage, resulting the largest amount of reducing sugars accumulated at 90 days of storage in gibberellin. In contrast, (*S*)-carvone significantly up-regulated the expression of *GBSS* at 60 days of storage, leading to significant difference in reducing sugars at 60 days of storage. *INH1/2* were up-regulated in the first 60 days of storage and showed significant difference in *INH2* at 60 days of storage, causing the inhibition of reducing sugars accumulation.

## 3. Discussion

As a major raw material of potato chips, “Atlantic” potato tubers have faced a great challenge in how to tackle with the problem of CIS during their storage. CIS can exert a negative influence on the quality of potato chips. The key problem is that CIS facilitates Maillard reactions during frying processes, which may cause a dark coloration on the potato surface. Acrylamide, which causes health problems in humans, may also be generated. In order to address this question, many measurements have been taken.

Gibberellin plays a crucial role in plants during development, such as modulating leaf expansion, stimulating seed germination and influencing flower development [24]. Previous studies suggested that endogenous gibberellin was regulated by light, cold temperature and other stresses during plant growth [25]. In *Arabidopsis* seeds, the expressions of cold-responsive genes were found to be consistent with the expressions of gibberellin-related genes, indicating that gibberellin played a key role in response to cold temperature [26]. In addition, the dormancy was decreased in the *GA 20-oxidase* gene overexpressing potato plants whereas the reverse phenotype was occurred in the antisense *GA 20-oxidase* gene potato plants, which demonstrated that gibberellin was significantly related to potato dormancy [27]. With the breaking of dormancy, potatoes started to sprout accompanied by soaring reducing sugars levels providing energy for growth [28]. Our results indicated that gibberellin treatment shortened potato tubers’ dormancy, promoted the growth of sprouts and accelerated the accumulation of sugars. There is a possibility that gibberellin breaks down starch mainly through a hydrolysis process rather than starch phosphorylation and it enhances the response to cold resistance by accumulating the reducing sugars in potato tuber.

(*S*)-carvone is a GRAS food additive, mainly isolated from essential oil of caraway seed (*Carum carvi* L.). According to previous studies, (*S*)-carvone plays a crucial role in suppressing sprouting via inhibiting the activity of 3-hydroxy-3-methylglutaryl coenzyme A reductase (HMGR), an important enzyme in the mevalonate pathway, at a post-translational level in either *Nicotiana tabacum* or potato tuber [29,30]. In addition, it has been reported that (*S*)-carvone possessed high antioxidant ability, and the ability to suppress the growth of bacteria and fungi [23]. From our results, although with a small quantity of (*S*)-carvone, the reducing sugar accumulation was inhibited and the sprouting condition in potatoes was delayed during postharvest storage, which showed the potential for application in the postharvest potato industry.

In potato tubers, total sugar concentrations are primarily constituted by reducing sugars (glucose and fructose) and non-reducing ones (sucrose) [31]. Unlike glucose and fructose, the role of sucrose in the browning of fried potato products is limited [32]. However, influencing effects might develop as it may function as a transitory balance in the starch degradation which further leads to the formation of its respective monomers due to the enzyme invertase [32]. According to our results, a closer association between sugar metabolism and percentage sprouting during postharvest potato storage was observed. Potato tubers with high levels of sprouting showed higher sugar concentrations as observed in the control and gibberellin-treated potatoes. A possible reason might be the degradation of starch into soluble sugars in sprouted potatoes which also been reported by Fauconnier et al. [33]. The presence of sugar in any form is of primary concern due to its possible participation in the Maillard reaction and subsequent acrylamide formation during industrial processing [34]. The postharvest interventions, either as sprout inhibitor or sprout suppressant maintained lower sugar contents as compared to the control during storage [32]. A similar observation was confirmed by the findings of Karanisa et al. [35] in carvone-treated potato tubers.

There are various of enzymes involved in the process of CIS. Degradation of starch mainly occurs via phosphorylation or hydrolysis pathways [36,37]. Overexpressing *BAM* isolated from *Poncirus trifoliate* in tobacco led to increase starch degradation rate and stimulate maltose and soluble sugars accumulation [38]. Antisense *BAM* expression was also reported to cause an ascend trend of starch accumulation in potato leaves when compared with wild plants, suggesting that BAM is a key enzyme in starch hydrolysis [39]. In addition, ectopic expression of a mutated *AGPase* gene from *E. coli* to potato tubers exhibited an increasing trend in starch contents, indicating that *AGPase* is responsible for the formation of starch [40]. It has been reported that the expression of *UGPase* was enhanced by salt stress in tomato leaves, causing the accumulation of sucrose to maintain the osmotic equilibrium and improve the plant resistance [41]. Ciereszko et al. cultivated *Arabidopsis thaliana* under cold and water stress separately and found that *UGPase* was up-regulated significantly after treatment, resulting in a strong accumulation of sucrose [42]. *INH* was responses to biotic and abiotic stress in plant growth, which can inhibit the generation of reducing sugars [43]. From our results, gibberellin treatment can significantly stimulate starch hydrolysis through *BAM1/2*, meanwhile inhibited the expressions of *AGPase* and *GBSS* to restrict starch resynthesis. In addition, *UGPase* was up-regulated significantly by gibberellin, leading to accumulation of sucrose. The expressions of *INH1/2* were suppressed significantly during storage, which promoted sucrose converts into reducing sugars. Whereas, (*S*)-carvone significantly accelerated the rate of starch re-synthesis through modulating *GBSS* expression and caused the accumulation of starch instead of sucrose. Meanwhile, (*S*)-carvone stimulated the expression of *INH1/2*, leading to a slow speed of reducing sugar accumulation in potatoes during low temperature storage.

## 4. Materials and Methods

### 4.1. Plant Material and Treatments

The research was conducted with tubers of “Atlantic” potato (*Solanum tuberosum* L.), which were bought from Linkage Inc. (Inner Mongolia, China). After seven days of wound curing under room temperature, the tubers were divided randomly into three groups: (a) Control: untreated potatoes; (b) Gibberellin treatment: potatoes were sprayed by 0.015 g L^−1^ gibberellin solutions three times in one day; (c) (*S*)-carvone treatment: potatoes were sprayed by 6.5 mL L^−1^ (*S*)-carvone solutions three times in one day. The concentrations of gibberellin solution and (*S*)-carvone solution were selected based on relevant studies, and both of them were dissolved in absolute ethanol. A continuous temperature of 4 °C and 95% relative humidity was maintained during the whole storage period. The potato tubers were put in the generally dark cold storage and no other light or gas was imposed into the zone. The tubers were sampled at 0 day, 30 days, 60 days and 90 days, respectively. Each sampling point consist of three biological replicates containing nine potatoes in each. The samples were shifted immediately to the laboratory, frozen with liquid nitrogen and stored at −80 °C for further analysis.

### 4.2. Sprouting Rate Analysis

Three replicates were considered to calculate sprouting rate, containing 45 potatoes in each treatment. Sprouting rate was calculated as follows: Sprouting rate (%) = *n*/*n*_0_ × 100%. In which, “*n*” represents the number of sprouting potatoes, “*n*_0_” represents the number of total potatoes in each replicate. The final result was represented by the average of three replicates.

### 4.3. Weight Loss Rate Analysis

The weight of each sample was recorded at each sampling point. The weight loss rate was calculated as follows: Weight loss rate (%) = (*m*_0_ − *m*)/*m*_0_ × 100%. In which, “*m*_0_” represents the initial weight of each sample, “*m*” represents the sample’s weight in each sampling point. The finial result was calculated by the average of the three replicates.

### 4.4. Respiration Rate Analysis

Each group of three potatoes were packed with a plastic bag and stored at 4 °C for further measurement. The gas constituents were measured by gas analyzer (checkmate 3, PBI-DanSenser, Ringsted, Denmark) at 0 day, 30 days, 60 days and 90 days of storage. The respiration rate was measured by means of percent CO_2_ produced per volume of the bag.

### 4.5. Sugar and Starch Content Analysis

The sugar content of each treatment was extracted and measured according to Lin et al. with some modifications [16]. In brief, 0.5 g of sample powder (dried for 3 days before use) was homogenized with 8 × 10^−3^ L of 80% ethanol. The extraction method was continued by ultra-sonication for 30 min at 50 °C, then centrifuged at 9500× *g* for 10 min. The supernatant was removed and used for sugars analysis. The precipitate was dissolved with 8 × 10^−3^ L of 80% ethanol, then centrifuged at 9500× *g* for 10 min. After that, the precipitate was dissolved with 8 × 10^−3^ L of ddH_2_O, then centrifuged at 9500× *g* for 10 min. The precipitate was transferred to beaker and incubated with ddH_2_O at 100 °C to make it clear. And then the solution was diluted to 100 mL for starch analysis. The starch was analyzed using ultraviolet spectrophotometry. Aliquots of 1 × 10^−3^ L of the supernatant were transferred into a new tube and dried with pure nitrogen gas. The same volume of ddH_2_O was added to dissolve the residue completely and the mixture filtered with a 0.22 µm membrane. 10 µL of each prepared sample were subjected to ion chromatography (ICS-3000, Dionex, Sunnyvale, CA, USA) using a Carbo PacTMPA20 column (3 mm × 150 mm). The following procedure was conducted at 35 °C, with a 0.5 mL min^−1^ flow rate. The gradient elution buffer was set as follows: A. ddH_2_O; B. 0.25 mol L^−1^ NaOH; Equal gradient of 92.5% A and 7.5% B were used for elution. A pulsed amperometric detector with a gold electrode was used. The sugars were identified via the retention times of standard compounds and calculated though comparison with standard curves. The content of total reducing sugars was reported as the sum of glucose and fructose.

### 4.6. RNA Extraction and cDNA Synthesis

Total RNA was extracted by Quick RNA isolation Kit (Huayueyang, Beijing, China) from 0.3 g of potato tissue. Using the kit of TransScript^®^ One-Step cDNA Removal and cDNA Synthesis SuperMix (TransGen Biotech, Beijing, China) with both Anchored Oligo (dT)_18_ and Random Primer (N9), the first strand cDNA was synthesized by reverse transcription by following the instruction.

### 4.7. Quantitative Real-Time PCR (qRT-PCR)

Quantitative real-time PCR was performed on an ABI 7500 instrument (Applied Biosystems, Thermo Fisher Scientific, Waltham, MA, USA) using TransStart Top Green qPCR SuperMix (+Dye II) (TransGen Biotech, Beijing, China). The reaction system of qRT-PCR was presented as Table 1. The relative gene expression was normalized with the reference gene *EF1a*. The program was initiated for 10 min at 95 °C, 15 s at 95 °C and 1 min at 60 °C, after which, there were 40 cycles of 95 °C for 15 s and 60 °C for 1 min, then completed with a melting curve analysis procedure. The primers were designed by Primer Premier Software 5.0 (PREMIER Biosoft International, Palo Alto, CA, USA) and the sequences were shown in Table 2. Three biological replicates consisting of fifteen tubers were considered for gene expression analysis for each sample and repeated three times for precision of results.

### 4.8. Statistical Analysis

All experiments were carried out in a completely randomized design and each treatment contained three replicates unless stated otherwise. The figures were drawn using Origin 8.6 software (Microbial So ware Inc., Northampton, MA, USA). The significant difference was calculated at the 0.05 level by SPSS Statistics 20 Software (IBM Corporation, Armonk, NY, USA).

## 5. Conclusions

From the above results and comparative discussions, it can be concluded that the application of gibberellin can significantly accelerate CIS while the opposite effect was found with (*S*)-carvone treatment, suggesting a primary function of *BAM1/2*, *AGPase*, *GBSS*, *UGPase* and *INH1/2* in the sugar metabolism in potato tubers during cold storage. Furthermore, the physiological properties of (*S*)-carvone-treated potato tubers were far greater than that in other groups of potato tubers, even with a slight concentration use, especially the sprouting condition, which is suppressed deeply during the later stages of storage. Thus, (*S*)-carvone can be considered as a potential sugar inhibitor. Nevertheless, the chemical concentration, treatment time, and also the treatment method need to be optimized before industrial application can be considered.

## Figures and Tables

**Figure 1 molecules-23-03118-f001:**
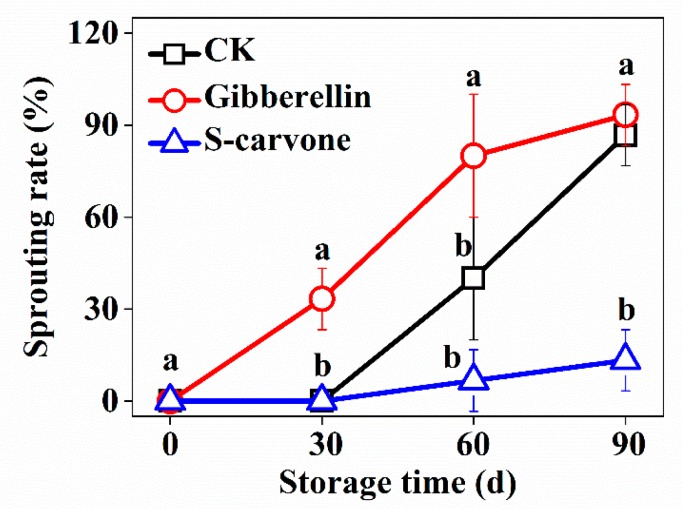
Effects of gibberellin and (*S*)-carvone on sprouting rate in potato tubers during 4 °C low temperature storage. Error bar represents level of significance (*p* < 0.05) among replicated samples (*n* = 45). Means with the same letters are not statistically different according to the least significant difference test at *p* < 0.05.

**Figure 2 molecules-23-03118-f002:**
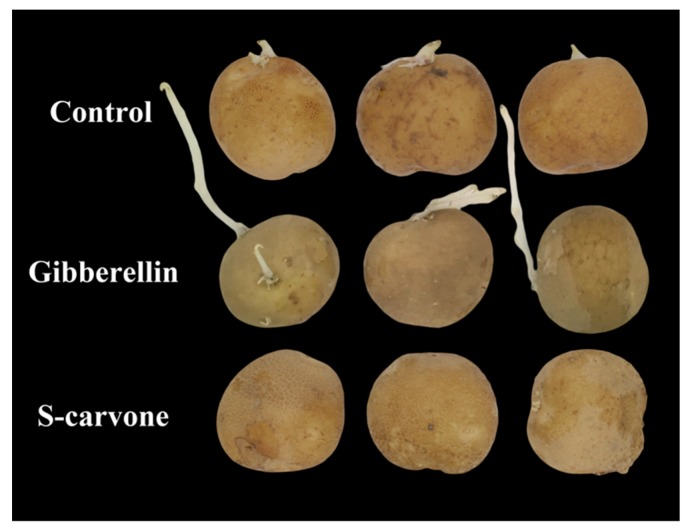
Effects of gibberellin and (*S*)-carvone on sprouting in potato tubers at 90 days of 4 °C low temperature storage.

**Figure 3 molecules-23-03118-f003:**
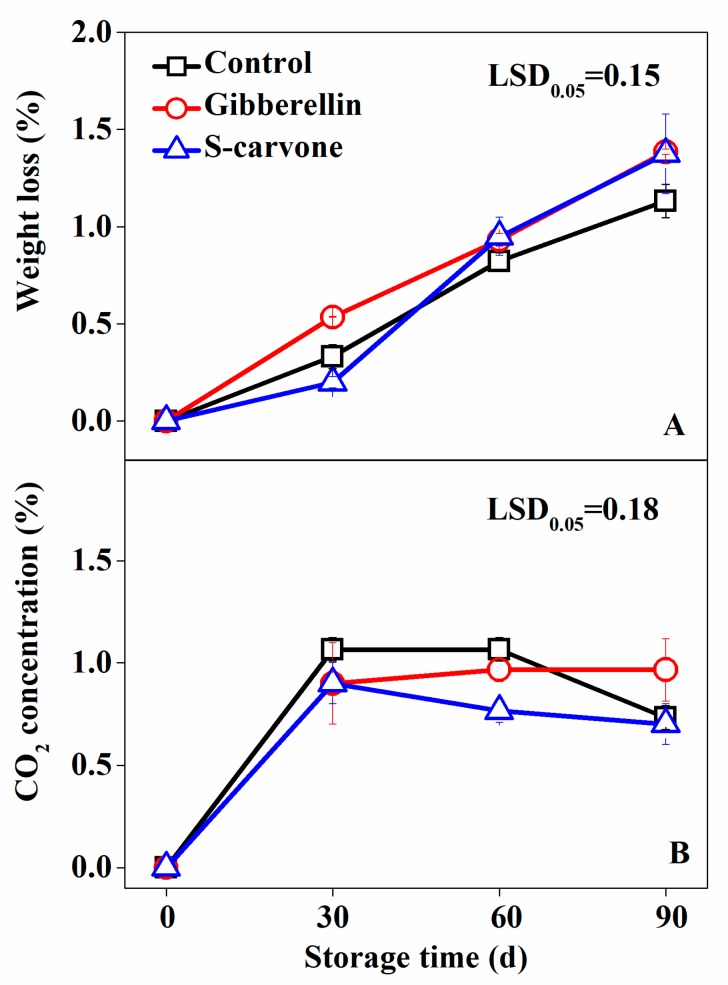
Effects of gibberellin and (*S*)-carvone on physiological response in potato tubers during 4 °C low temperature storage. (**A**) Gibberellin and (*S*)-carvone effect on weight loss. (**B**) Gibberellin and (*S*)-carvone effect on CO_2_ concentration. The error bars represent the standard errors. LSDs represent least significant differences at the 0.05 level.

**Figure 4 molecules-23-03118-f004:**
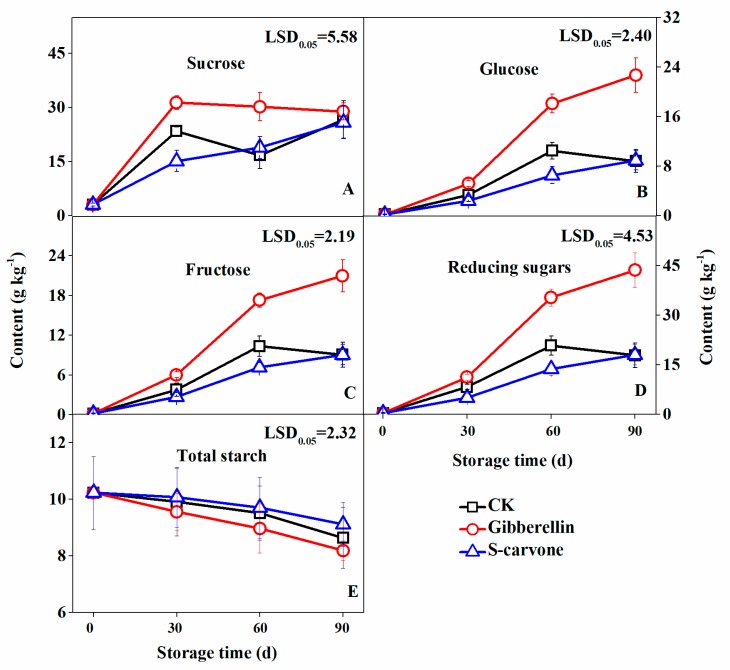
Effects of gibberellin and (*S*)-carvone on sugars contents in potato tubers during 4 °C low temperature storage. (**A**) Gibberellin and (*S*)-carvone effect on sucrose content. (**B**) Gibberellin and (*S*)-carvone effect on glucose content. (**C**) Gibberellin and (*S*)-carvone effect on fructose content. (**D**) Gibberellin and (*S*)-carvone effect on reducing sugars content. (**E**) Gibberellin and (*S*)-carvone effect on total starch content. The error bars represent the standard errors. LSDs represent least significant differences at the 0.05 level.

**Figure 5 molecules-23-03118-f005:**
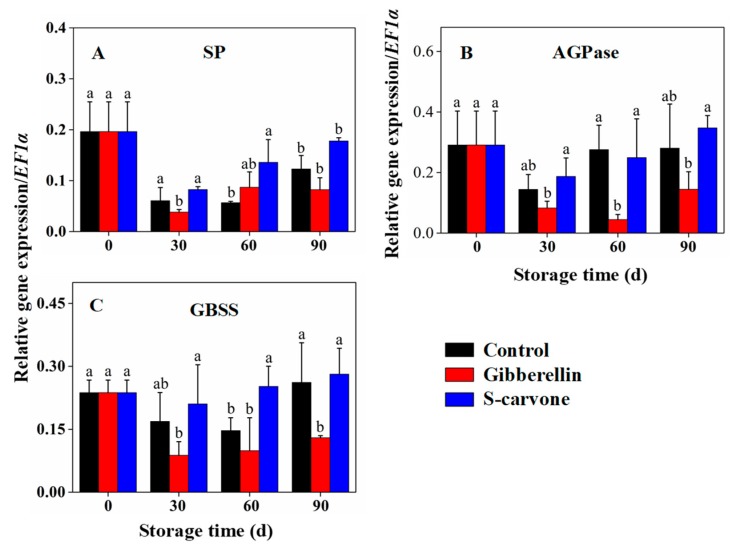
Effects of gibberellin and (*S*)-carvone on gene expression/*EF1a* in potato tubers during 4 °C low temperature storage. (**A**) Gibberellin and (*S*)-carvone effect on expression of *SP*. (**B**) Gibberellin and (*S*)-carvone effect on expression of *AGPase*. (**C**) Gibberellin and (*S*)-carvone effect on expression of *GBSS*. The error bars represent the standard errors. Means with the same letters are not statistically different according to the least significant difference test at *p* < 0.05.

**Figure 6 molecules-23-03118-f006:**
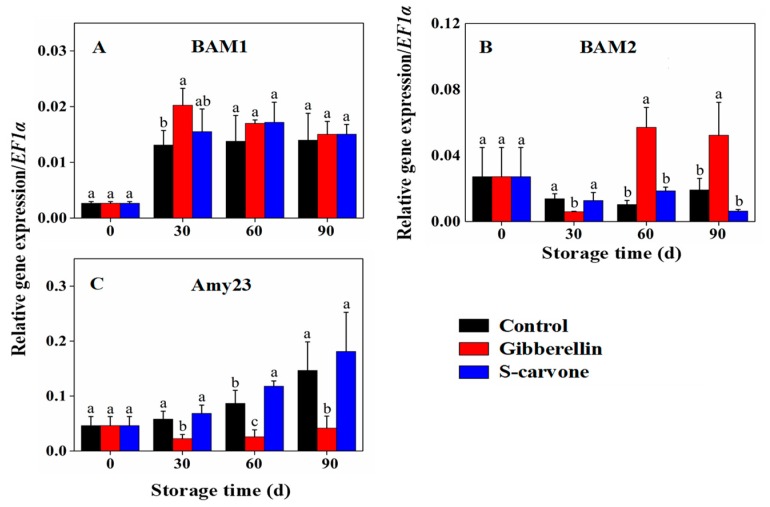
Effects of gibberellin and (*S*)-carvone on gene expression/*EF1a* in potato tubers during 4 °C low temperature storage. (**A**) Gibberellin and (*S*)-carvone effect on expression of *BAM1*. (**B**) Gibberellin and (*S*)-carvone effect on expression of *BAM2*. (**C**) Gibberellin and (*S*)-carvone effect on expression of *Amy23*. The error bars represent the standard errors. Means with the same letters are not statistically different according to the least significant difference test at *p* < 0.05.

**Figure 7 molecules-23-03118-f007:**
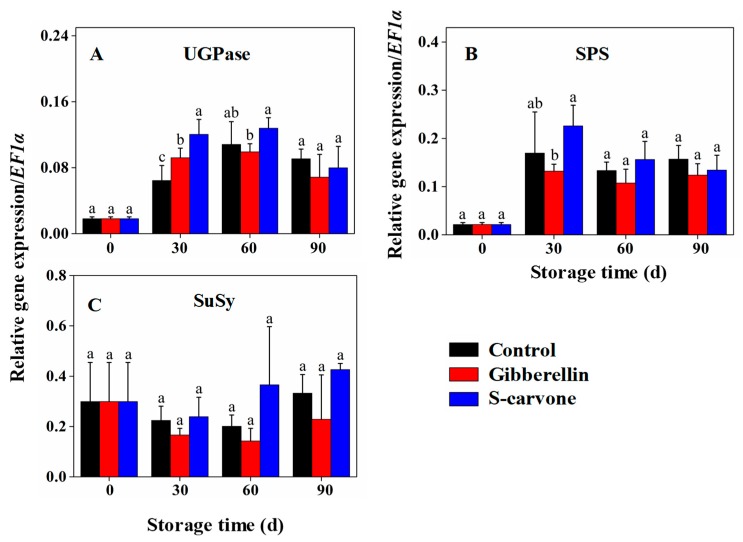
Effects of gibberellin and (*S*)-carvone on gene expression/*EF1a* in potato tubers during 4°C low temperature storage. (**A**) Gibberellin and (*S*)-carvone effect on expression of *UGPase*. (**B**) Gibberellin and (*S*)-carvone effect on expression of *SPS*. (**C**) Gibberellin and (*S*)-carvone effect on expression of *SuSy*. The error bars represent the standard errors. Means with the same letters are not statistically different according to the least significant difference test at *p* < 0.05.

**Figure 8 molecules-23-03118-f008:**
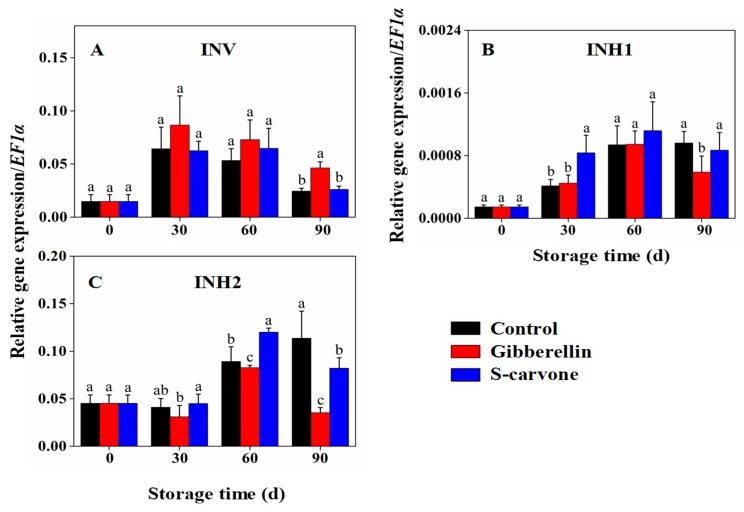
Effects of gibberellin and (*S*)-carvone on gene expression/*EF1a* in potato tubers during 4°C low temperature storage. (**A**) Gibberellin and (*S*)-carvone effect on expression of *INV*. (**B**) Gibberellin and (*S*)-carvone effect on expression of *INH1*. (**C**) Gibberellin and (*S*)-carvone effect on expression of *INH2*. The error bars represent the standard errors. Means with the same letters are not statistically different according to the least significant difference test at *p* < 0.05.

**Table 1 molecules-23-03118-t001:** Reaction system of qRT-PCR.

Reagent	Volume/μL
Template	1
Forward Primer (10 μM)	0.4
Reverse Primer (10 μM)	0.4
2× TransStart® Top Green qPCR SuperMix (+Dye II)	10
ddH_2_O	8.2

**Table 2 molecules-23-03118-t002:** Primer sequences used in this research.

Gene Name	Gene ID	Forward Primer	Reverse Primer
*EF1a*	PGSC0003DMT400014674	ATTGGAAACGGATATGCTCCA	TCCTTACCTGAACGCCTGTCA
*SP*	PGSC0003DMT400008970	CAGGAACCAGATGCTGCTCTT	CATAGCCCCATGCTGGGTAGT
*AGPase*	PGSC0003DMT400079823	GGAGTCCGATTCAATGTGAGAAGAAG	CCAAAACACTCCGGCTAGCATC
*GBSS*	PGSC0003DMT400031568	TACACAAGAGTGGAACCCAGCGAC	TGTCAACAGGCAAGCCAACTGC
*UGPase*	PGSC0003DMT400034699	CCATCGAGTTGGGACCTGAA	GGGAATAGACTTGAAACGGCCTAAG
*SPS*	PGSC0003DMT400067951	TCCACAGGTCGCAAGAGTATCAGG	CCGGATAAAACACTTCGCTCCCAC
*SuSy*	PGSC0003DMT400007506	TTTGAGGCCTGGTGTCTGGGAATACA	TCCATTCGAGGCTCCGTCGACAA
*INH1*	PGSC0003DMT400004650	GCAAGGTTCGGTAGGTATGA	AAACAGTGAGGGTATTGGGT
*INH2*	PGSC0003DMT400011760	ACTATCAAAAGTTGCTAAATCC	CCCTTCTTCAAACCTCGTAT
*INV*	PGSC0003DMT400035987	CAGGGTCTAGCGTGACTGC	TGATGGGACATCGGTGAAA
*BAM1*	PGSC0003DMT400003933	TGAGATGCGTGACCATGAGC	CAAGTGGAACTTGCGCTTCC
*BAM2*	PGSC0003DMT400004686	AGCACGTATGTTAGCGAAACA	GTTGAACTAAGCCTTCTGGTGA
*Amy23*	PGSC0003DMT400025601	GGCATACACAGCCGTTCATCT	ATCCGTCCCCAATCTTCACG

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
