# Peer review of "Effects of (S)-Carvone and Gibberellin on Sugar Accumulation in Potatoes during Low Temperature Storage"

_molecules, 2018, doi:10.3390/molecules23123118_

Reviewer 1 Report

The authors have made attempts to answer the previous comments by reviewers and have made some improvements. Unfortunately, there are still a number of issues with this work that make it unacceptable for publication in my opinion.

I agree with one of the other reviewers that the present results do not provide enough evidence to sustain the argument that s-carvone inhibits potato sprouting through the regulation of starch and sugar metabolism. There are some changes but the number of significant differences between treatments are few. The role of GAs in potato sprouting has already been studied and so the data presented regarding their effects in this paper is largely not novel. It seems that this manuscript is a preliminary report that needs further experimental study to prove the case put forward. In the reply to reviewers the authors themselves said ‘The results of the s-carvone treatment were not strong enough, but it did show some effect on inhibiting sugar accumulation’. It would seem that it would be better if the further studies they suggest should be completed and used to produce a much stronger and more important paper than the one currently submitted. The authors may do themselves a disservice by prematurely publishing this work.

It should also be noted that the authors are discussing changes in gene expression, not enzyme activities, and relating them to metabolite levels. Changes in gene expression do not necessarily lead to changes in protein level and metabolite level. There may be a correlation between gene expression level and a change in sugars but it is only a correlation as there is no chain of proof of cause and effect. As an example, the final paragraph of the Discussion has comments (lines 301-309) that assume that changes in gene expression are responsible for the observed changes in carbohydrates. These statements are too speculative and are unsupported.

There are still errors in the interpretation of the data, see below.   

I understand that English is not the first language of the authors and that they have made a significant effort to write their text clearly but there are still many cases where the words used and the structure of the sentence either obscure the proper meaning or are not consistent with the data. I do not think that it is the job of reviewers to correct all of these issues. Lines 403-406 are one example but there are many others. There is evidence from other papers that s-carvone can affect sugar metabolism therefore, this is not a novel finding.

There are papers not quoted by the authors that seem directly relevant to this topic (GA, s-carvone treatment of potatoes under cold storage) that are not quoted and would add further to the discussion. Some of these are more relevant than the general papers discussing the role of GAs in plants currently used in the introduction and discussion.

 Other issues:

Lines 179-181 make absolutely no sense if they are referring to Figure 3D. Do the authors mean to refer to ‘total starch content’ as presented in Figure 3E??

Lines 209-210 the statement that s-carvone treatment did not affect the expression of ‘these genes significantly’ is not correct as the expression of Amy23 at 60d was higher in s-carvone-treated potatoes than in the control.

Line 222, 223 there was no significant difference in the expression of SPS in the Control and GA-treated samples at 30 d of storage.

Line 236 change ‘CK’ to ‘control’.

Line 236 INV expression was significantly higher in GA-treated potatoes at 90 d.

Line 238-240 this needs to be made clear. There was a sig. difference at 30 d for INH1 and at 60 d for INH2, INH2 was sig. lower than the control at 90 d.

Line 248 BAM1 expression was only upregulated at 30 d, there are no data points between 0 and 30 d.

Line 251 GA down-regulated AGase at 60 d, GBSS at 90 d, please be precise.

Author Response

Comments and Suggestions for Authors The authors have made attempts to answer the previous comments by reviewers and have made some improvements. Unfortunately, there are still a number of issues with this work that make it unacceptable for publication in my opinion. I agree with one of the other reviewers that the present results do not provide enough evidence to sustain the argument that s-carvone inhibits potato sprouting through the regulation of starch and sugar metabolism. There are some changes but the number of significant differences between treatments are few. The role of GAs in potato sprouting has already been studied and so the data presented regarding their effects in this paper is largely not novel. It seems that this manuscript is a preliminary report that needs further experimental study to prove the case put forward. In the reply to reviewers the authors themselves said ‘The results of the s-carvone treatment were not strong enough, but it did show some effect on inhibiting sugar accumulation’. It would seem that it would be better if the further studies they suggest should be completed and used to produce a much stronger and more important paper than the one currently submitted. The authors may do themselves a disservice by prematurely publishing this work. It should also be noted that the authors are discussing changes in gene expression, not enzyme activities, and relating them to metabolite levels. Changes in gene expression do not necessarily lead to changes in protein level and metabolite level. There may be a correlation between gene expression level and a change in sugars but it is only a correlation as there is no chain of proof of cause and effect. As an example, the final paragraph of the Discussion has comments (lines 301-309) that assume that changes in gene expression are responsible for the observed changes in carbohydrates. These statements are too speculative and are unsupported.

Reply: We are agreed with the reviewer comments. Hence, we have deleted the information unsupported by our results or literatures. There are still errors in the interpretation of the data, see below.  

I understand that English is not the first language of the authors and that they have made a significant effort to write their text clearly but there are still many cases where the words used and the structure of the sentence either obscure the proper meaning or are not consistent with the data. I do not think that it is the job of reviewers to correct all of these issues. Lines 403-406 are one example but there are many others. There is evidence from other papers that s-carvone can affect sugar metabolism therefore, this is not a novel finding.

Reply: We are very sorry as this kind of misleading of our writing. We have deleted the line unsupported by a real evidence.

There are papers not quoted by the authors that seem directly relevant to this topic (GA, s-carvone treatment of potatoes under cold storage) that are not quoted and would add further to the discussion. Some of these are more relevant than the general papers discussing the role of GAs in plants currently used in the introduction and discussion.

Reply: We are sorry for our shortcoming in this part. We have added some related study in line 235-248.

Other issues: Lines 179-181 make absolutely no sense if they are referring to Figure 3D. Do the authors mean to refer to ‘total starch content’ as presented in Figure 3E??

Reply: We are sorry for the mislead of our writing. We have made correction in line 223-224

Lines 209-210 the statement that s-carvone treatment did not affect the expression of ‘these genes significantly’ is not correct as the expression of Amy23 at 60d was higher in s-carvone-treated potatoes than in the control.

Reply: We have revised it in line 151-152.

Line 222, 223 there was no significant difference in the expression of SPS in the Control and GA-treated samples at 30 d of storage.

Reply: We have revised the related expression in line 163-164.

Line 236 change ‘CK’ to ‘control’.

Reply: We have revised it in line 178.

Line 236 INV expression was significantly higher in GA-treated potatoes at 90 d.

Reply: We have cleared it in line 177-179. Line 238-240 this needs to be made clear.

There was a sig. difference at 30 d for INH1 and at 60 d for INH2, INH2 was sig. lower than the control at 90 d.

Reply: We have cleared it in line 182-183.

Line 248 BAM1 expression was only upregulated at 30 d, there are no data points between 0 and 30 d.

Reply: We have revised it in line 193.

Line 251 GA down-regulated AGase at 60 d, GBSS at 90 d, please be precise.

Reply: We have revised it in line 134-135.

Reviewer 2 Report

At present manuscript, the authors would like to clarify the effects of GA and s-carvone on sugar accumulation during low temperature storage in potatoes. They have revised all the concern and provide enough evidence to support their surmise. The English writing is fine. Only minor revise is necessary. The detail concerns as followed:

1.     Line 72, the title of Table 1, NO space is necessary between 4. Please check the format whole manuscript.

2.     Line 74, should revise Note: ……. were significant differences.

3.     The title of Y axis in Figure 4 to Figure 7 need to put the suitable positions..

4.     The format of references list are still inconsistent, Please check all the references and follow the Guide line to Authors.

Author Response

At present manuscript, the authors would like to clarify the effects of GA and s-carvone on sugar accumulation during low temperature storage in potatoes. They have revised all the concern and provide enough evidence to support their surmise. The English writing is fine. Only minor revise is necessary. The detail concerns as followed: 

1.     Line 72, the title of Table 1, NO space is necessary between 4. Please check the format whole manuscript.

Reply: Thanks for your kind suggestion. We have deleted all the spaces between number and °C.

2.     Line 74, should revise Note: ……. were significant differences.

Reply: Thanks for your kind suggestion. We have changed this table into chart according to another reviewer’s suggestion (Line 82).

3.     The title of Y axis in Figure 4 to Figure 7 need to put the suitable positions.

Reply: We have changed the figures as other reviewer suggested too.

4.     The format of references list are still inconsistent, Please check all the references and follow the Guide line to Authors.

Reply: Thanks for your suggestion. We have revised the references according to Guide line to Authors.

Reviewer 3 Report

Dear Authors,

I had great honor to review Manuscript ID: molecules-386472, entitled: “Effects of s-carvone and gibberellin on sugar accumulation during low temperature storage in potatoes” which is considered for publication in Journal: Molecules. This manuscript presents very interesting and vital results in context of potato storage in low temperatures. Problem of potato storage is worldwide problem which in some regions (less developed) transform into decision death or alive in context of hunger. However, authors of manuscript in my opinion make some shortcomings in particular section of manuscript which is very problematic. Therefore, I would like present list of my critical remarks to authors.

Abstract section:

Generally well written. Only one mistake (mental shortcut) in line 25 “…potential sugar inhibitor…”. This is an error because s-carvone as author presents is inhibitor of gene expression of some enzymes not all sugars in potato.

Introduction sections

In my opinion this part is too short to introduce in appropriate way research hypothesis. I would like suggest to add more information about parameters which defines potato tuber dormancy in general. Because in results section authors starts with time of dormancy ending. All information about CIS in this section is generally ok but authors forgot that many spotted CIS is effect of viral infection (different types of plant viruses) in tubers which also generates dark color inside tube and generation of accumulate reducing sugars during storage.

In line 36 acrylamide is carcinogenic “not probably”  especially in context acrylamide resins.

Results section

Line 72: Table 1 will be more interesting if results will be presented on a chart with statistic. So I would like to suggest to change table to figure.

Line 111-153: Text of results and Figures 4,5 and 6 description should be little changed. Firstly if on Figure chart is letter A,B or C then in text of results Figures should be cited for example “…expression of SP (Figure 4A)”... Moreover, if on figures are letters A,B…etc. then description should be presented like that:

“…Figure 4. Effects of gibberellin and s-carvone on gene expression. A. Gibberellin and s-carvone  effect on expression of SP. B. Gibberellin and s-carvone  effect on expression of AGPase” and so on

All charts on figures 4,5,6 should be separated because in some cases y axis are connected.

Line 156-166: Some fragments should be transferred to discussion because it fits there better

For example: “…From the above results, gibberellin treatment breaks down starch mainly through hydrolysis process rather than starch phosphorylation…”

Discussion section

For amount of presented result this section is too short. I would like to suggest to enlarge discussion.

Material and Methods

Authors should add more information about tuber storage conditions, atmosphere composition, temperature light or darkness conditions???

Line 211: How much tubers was tested for expression also 45 from each treatment???

Other comments/questions to the article:

I would like outlined is commonly known fact that s-carvone is harmful for human health and can also penetrate to parenchyma cells in plants. So it is question how long after treatment it residues in tubers. Because, if we cannot eliminated it from tubers then this process may be problematic even for further tests.

More problematic is that authors used only one variety for this research and this is little inadequate. To different varieties is minimal limit for such research about dormancy. Because potato varieties are highly varied each other even in level of endogenous gibberellin. So reaction of potato varieties may be different.

Last elements is that authors did not add any information about health of plant material. All plant material used to understanding of blocking CIS or sprouting must be first checked if is free from plant pathogens with can change reaction of tubers.

Because all of my comments from above I would like suggest major reviesion

Author Response

Comments and Suggestions for Authors

Dear Authors,

I had great honor to review Manuscript ID: molecules-386472, entitled: “Effects of s-carvone and gibberellin on sugar accumulation during low temperature storage in potatoes” which is considered for publication in Journal: Molecules. This manuscript presents very interesting and vital results in context of potato storage in low temperatures. Problem of potato storage is worldwide problem which in some regions (less developed) transform into decision death or alive in context of hunger. However, authors of manuscript in my opinion make some shortcomings in particular section of manuscript which is very problematic. Therefore, I would like present list of my critical remarks to authors.

Abstract section:

Generally well written. Only one mistake (mental shortcut) in line 25 “…potential sugar inhibitor…”. This is an error because s-carvone as author presents is inhibitor of gene expression of some enzymes not all sugars in potato.

Reply: Thanks for your kind suggestion. We have revised this expression into a clear way in line 25. The meaning of the sentence we wanted to express was that through the results (s-carvone is inhibitor of gene expressions of some CIS related enzymes), it can be speculated that s-carvone can work in decreasing sugars and alleviating CIS.

Introduction sections 

In my opinion this part is too short to introduce in appropriate way research hypothesis. I would like suggest to add more information about parameters which defines potato tuber dormancy in general. Because in results section authors starts with time of dormancy ending. All information about CIS in this section is generally ok but authors forgot that many spotted CIS is effect of viral infection (different types of plant viruses) in tubers which also generates dark color inside tube and generation of accumulate reducing sugars during storage. 

Reply: Thanks for your kind suggestions. We have added some information related to potato dormancy and factors affecting this in line 33-41. Microbial influence on CIS was not considered in this study and the potato tuber was collected from disease free plants and seed potato was treated externally to mitigate seed borne pathogen attack. However, reviewer noted a very important parameter which would be considered in further study.

In line 36 acrylamide is carcinogenic “not probably” especially in context acrylamide resins.

Reply: Thanks for your kind suggestion. We have deleted the “probably” in Line 45.

Results section

Line 72: Table 1 will be more interesting if results will be presented on a chart with statistic. So I would like to suggest to change table to figure.

Reply: Thanks for your kind suggestion. We have changed the table to line chart in line 82 (Figure 1.).

Line 111-153: Text of results and Figures 4,5 and 6 description should be little changed. Firstly if on Figure chart is letter A,B or C then in text of results Figures should be cited for example “…expression of SP (Figure 4A)”... Moreover, if on figures are letters A,B…etc. then description should be presented like that:

“…Figure 4. Effects of gibberellin and s-carvone on gene expression. A. Gibberellin and s-carvone  effect on expression of SPB. Gibberellin and s-carvone  effect on expression of AGPase” and so on

Reply: Thanks for your suggestion. We have corrected the format of Figures 3-8 as you presented above.

All charts on figures 4,5,6 should be separated because in some cases y axis are connected.

Reply: We have changed figures 4,5,6,7 according to reviewer suggestion (line-144, 154, 170 185).

Line 156-166: Some fragments should be transferred to discussion because it fits there better 

For example: “…From the above results, gibberellin treatment breaks down starch mainly through hydrolysis process rather than starch phosphorylation…”

Reply: Thanks for your kind suggestion. We have transferred it to the discussion (Line 223-224).

Discussion section

For amount of presented result this section is too short. I would like to suggest to enlarge discussion.

Reply: We have added some information in the part of the discussion (line 235-248).

Material and Methods 

Authors should add more information about tuber storage conditions, atmosphere composition, temperature light or darkness conditions???

Reply: Thanks for your kind suggestion. We have added some information in this part (line 279-281).

Line 211: How much tubers was tested for expression also 45 from each treatment???

Reply: 45 was the numbers of potato tubers used for analyzing sprouting rate which needs a pretty large quantity to make sure a relatively precise result. In gene expression measurement, we used three replications (fifteen potatoes) per treatment per sampling date and each gene expression experiment was repeated three times. Line 331-332.

Other comments/questions to the article:

I would like outlined is commonly known fact that s-carvone is harmful to human health and can also penetrate to parenchyma cells in plants. So it is question how long after treatment it residues in tubers. Because, if we cannot eliminated it from tubers then this process may be problematic even for further tests.

Reply: We are very sorry for our negligence in this part. Although we haven’t tested how long residual effects of s-carvone, we have selected the dose from previous studies. The s-carvone is belonged to GRAS (Generally recognized as safe) compounds and most of the cases literature didn’t mention  health hazardous effect after application. In addition, in some country as Netherlands it has been used commercially in tuber storage.

More problematic is that authors used only one variety for this research and this is little inadequate. To different varieties is minimal limit for such research about dormancy. Because potato varieties are highly varied each other even in level of endogenous gibberellin. So reaction of potato varieties may be different.

Reply: We are sorry for the issue noted by the reviewer. Thanks for your kind suggestion. We have selected the cultivar on the basis of CIS severity. Hence, we picked “Atlantic” potato tubers because they are commonly used in potato chips industry which would deeply influence by CIS. However, further experiment would be conducted including different cultivar to make appropriate use for commercial application on potato storage.

Last elements is that authors did not add any information about health of plant material. All plant material used to understanding of blocking CIS or sprouting must be first checked if is free from plant pathogens with can change reaction of tubers.

Reply:Thanks reviewer for his constructive review. The plant was grown in a commercial potato farm. All the cultural practices as irrigation, soil condition and the seed tuber were maintained by the farm management where seed tuber was treated to mitigate seed borne pathogens before planting. Additionally, we have collected the samples from disease and any external symptom free plants.

Because all of my comments from above I would like suggest major reviesion

Round  2

Reviewer 1 Report

The authors have made a number of changes to the comments of the referees and have corrected some inaccuracies (note 'rowing' in Line 34 should be 'growing'). More useful information had been provided in the Introduction.

Reviewer 3 Report

Dear Authors,

Dear Authors,

Almost all my suggestion was consider by authors. Therefore, I recommend this manuscript to publication in Molecules. I would like only outlined that authors for further studies should remember to use more than one varieties in comparison. And all plant material should be firstly tested for presence or absence of pathogens before any research associated with storage will be conducted.

Best regards